# Vegetation modulates the impact of climate extremes on gross primary production

Milan Flach[1,2,7], Alexander Brenning[2], Fabian Gans[1], Markus Reichstein[1,3], Sebastian Sippel[4,5], and Miguel D. Mahecha[1,3,6]

[1]Max Planck Institute for Biogeochemistry, Department Biogeochemical Integration, P. O. Box 10 01 64, D-07701 Jena, Germany
[2]Friedrich Schiller University Jena, Department of Geography, Jena, Germany
[3]German Centre for Integrative Biodiversity Research (iDiv), Leipzig, Germany
[4]Norwegian Institute of Bioeconomy Research, Ås, Norway
[5]ETH Zürich, Institute for Atmospheric and Climate Science, Switzerland
[6]University of Leipzig, Remote Sensing Center for Earth System Research, Germany
[7]now at: INWT Statistics GmbH, Berlin, Germany

**Correspondence:** Milan Flach (milan.flach@bgc-jena.mpg.de)

**Abstract.** Drought and heat events affect the uptake and sequestration of carbon in terrestrial ecosystems. Factors such as the duration, timing and intensity of extreme events influence the magnitude of impacts on ecosystem processes such as gross primary production (GPP), i.e. the ecosystem uptake of $CO_2$. Preceding soil moisture depletion may exacerbate these impacts. However, some vegetation types may be more resilient to climate extremes than others. This effect is insufficiently understood
at the global scale and is the focus of this study. Using a global upscaled product of GPP that scales up in-situ land $CO_2$ flux observations with global satellite remote sensing, we study the impact of climate extremes at the global scale. We find that GPP in grasslands and agricultural areas is generally reduced during heat and drought events. However, we also find that forests, if considered globally, appear not in general to be particularly sensitive to droughts and heat events that occurred during the analyzed period or even show increased GPP values during these events. On the one hand, normal to increased GPP values are
in many cases plausible, e.g. when conditions prior to the event have been particularly positive. On the other hand, however, normal to increased GPP values in forests may also reflect a lack of sensitivity in current remote sensing derived GPP products to the effects of droughts and heatwaves. The overall picture calls for a differentiated consideration of different land cover types in the assessments of risks of climate extremes for ecosystem functioning.

## 1   Introduction

We expect that climate change leads to increases in frequencies, durations, intensities, and spatial extents of droughts and heatwaves in the next decades (Meehl et al., 2000; Olesen and Bindi, 2002; Seneviratne et al., 2012; Coumou and Robinson, 2013; Cook et al., 2015; Zscheischler and Seneviratne, 2017). Ecosystems will respond to the events ahead in multiple ways. In particular the processes controlling the terrestrial carbon balance, i.e. photosynthesis and respiratory processes as well as fires and e.g. pest-induced mortality are expected to be affected (Peuelas et al., 2004; Ciais et al., 2005; Vetter et al., 2008; Reichstein

et al., 2013; Bastos et al., 2014; Yoshida et al., 2015; Wolf et al., 2016; Brando et al., 2019) (for a recent review see Sippel et al. (2018)). Given that these responses represent feedbacks to the coupled climate–ecosystem dynamics, it is important to understand which factors generally influence the magnitudes of such impacts at the global scale (Frank et al., 2015). Previous studies have shown that event duration can be as important as intensity in controlling the reduction of gross primary production (GPP), which represents the total ecosystem carbon uptake (Granier et al., 2008; von Buttlar et al., 2018; Orth and Destouni, 2018). In particular, compound extreme events, e.g., the combination of drought and heat stress can increase the impact on GPP as compared to singular stressors (Ciais et al., 2005; AghaKouchak et al., 2014; Zscheischler et al., 2018; von Buttlar et al., 2018). Several case studies point to the crucial role of timing of the extreme event in influencing the magnitude of impacts on ecosystem functioning. Warm and early springs may partly compensate for severe carbon impacts of summer droughts (Wolf et al., 2016). In contrast, soil moisture depletion in spring can even enhance carbon losses during summer (Buermann et al., 2013; Sippel et al., 2017a; Buermann et al., 2018).

One important aspect is the question how strongly land cover types modulate drought and heat impacts on the fundamental processes controlling ecosystems carbon dynamics, such as gross primary production, ecosystem respiration, and net ecosystem exchange. Evidence from eddy covariance stations (von Buttlar et al., 2018) and case studies using spatiotemporal remote sensing derived data (Wolf et al., 2016; Flach et al., 2018) suggest that certain ecosystems are less vulnerable to heat and drought events than others. However, the question to what degree land cover types shape the impacts of droughts and heatwaves globally remains unclear. Here we aim to specifically investigate the importance of land cover type in controlling the impacts of climate extremes relative to other factors such as duration and magnitude of the extreme event.

When discussing impacts of climate extremes, the crucial question is their definition. One option is to use values over some global thresholds to detect extremes e.g. to detect temperatures above 25 or 30 degree Celsius and to investigate the associated anomaly in vegetation productivity. Another option is to define extreme events relative to some locally varying threshold, e.g. defined by the 95th percentile of the distribution of the data. Here, we rely on the latter definition, and refine the definition by taking also a joint multivariate distribution of the data with regionally varying thresholds into account (Flach et al., 2017, 2018). Furthermore, we restrict our analysis to those events that can also be considered a relative drought and heat event. We estimate anomalies regionally i.e. defining extreme events relative to the typical conditions of the regional growing season. We apply this method jointly to air temperature, surface moisture, and incoming shortwave radiation as fundamental variables to detect relative extreme events. Each event describes a spatiotemporal context that can be described by its spatial extent and duration (Zscheischler et al., 2013; Mahecha et al., 2017). The impacts are then assessed in these areas as anomalies in gross primary production (GPP). Our study addresses the impacts in the time range between 2003 and 2018 globally in different land cover classes and builds on nonlinear predictive models to understand the importance of the driving factors (for details see Methods, Section 2).

In the following, we will first start with the Methods (Section 2), including a subsection on the data, the preprocessing, the methods used for anomaly detection, the subsequent detection of spatio-temporally connected extreme events, and finally the statistical model to infer the main drivers of the GPP response during droughts and heatwaves. In the Results (Section 3), we will first show more generally the associated productivity during droughts and heatwaves in forest ecosystems and agricultural

systems. Then, we will explain the observed responses, first with a simple graphical approach, and then we will quantify the drivers of the observed responses with a statistical model. In the Discussion (Section 4), we will first elaborate on other studies, which found contrasting responses to climate extremes, and will then show how our findings can be interpreted (with a specific focus on forest ecosystems). Finally, we discuss potential biases and limitations of our approach and of the data used and finish with some Conclusions (Section 5).

## 2  Methods

For detecting hydrometeorological extreme events across ecosystems we need (i) a set of variables describing hydrometeorological extreme events and their impacts on productivity (Section 2.1), (ii) a detection algorithm (Section 2.2), and (iii) an approach to evaluate the hydrometeorological extremes with regard to responses in different ecosystems (Section 2.4).

**Table 1.** Grouping of the different ecosystems in the categories forest, agriculture and other.

| Land Cover Class | Category |
| --- | --- |
| Mixed Forest | Forest |
| Deciduous Broadleaf Forest | Forest |
| Evergreen Needleleaf Forest | Forest |
| Deciduous Needleleaf Forest | Forest |
| Evergreen Broadleaf Forest | Forest |
| Woody Savannas | Other |
| Savannas | Other |
| Grasslands | Other |
| C3 Cropland / Natural vegetation mosaic | Agriculture |
| C3 Croplands | Agriculture |
| C4 Fraction Cropland / Natural vegetation mosaic | Agriculture |
| C4 Fraction Croplands | Agriculture |
| Open Shrublands | Other |
| Close Shrublands | Other |
| Permanent Wetlands | Other |
| Urban and built-up | Other |

### 2.1  Data

To detect hydrometeorological extreme events we use 2-m air temperature, incoming shortwave radiation (both from ERA5, original resolution 0.25°, Copernicus Climate Change Service (C3S) (2017)), and surface moisture (v3.3b, original resolution 0.25° from the Global Land Evaporation Amsterdam Model (GLEAM) framework, (Miralles et al., 2011; Martens et al., 2017)). We consider surface moisture as a hydrometeorological variable due to its importance for drought detection although

it is influenced by vegetation. The impacts of the identified extremes are quantified as anomalies in gross primary productivity (GPP, original resolution $\frac{1}{12}^{\circ}$ from the remote sensing driven Fluxcom product (FLUXCOM-RS) Tramontana et al. (2016)). Anomalies in GPP are computed as deviations from the mean seasonal cycle excluding the extreme year itself. The selected hydrometeorological variables have global coverage and a common spatial resolution of 0.25°, and are used at an eight-daily temporal resolution covering the 2003–2018 period. The time period is chosen as it represents the common period of all data sets used (at the time of the analysis). Land cover classes at $\frac{1}{12}^{\circ}$ resolution (from the year 2010) were obtained from the Moderate Resolution Imaging Spectroradiometer (MODIS, collection 5, Friedl et al. (2010)) We group the available land cover classes in forest ecosystems (land cover classes containing "forest"), agricultural ecosystems (containing "crop"), and, all remaining other land cover types (Table 1).

## 2.2 Preprocessing and anomaly detection

We compute deviations from a smoothed median seasonal cycle in the hydrometeorological variables, which we denote as anomalies. For detecting extreme events, we apply a multivariate anomaly detection procedure described in detail in (Flach et al., 2018). It (i) accounts for seasonal changes in the variance of the anomalies using a moving window technique, and (ii) uses climatic similarities to obtain more robust thresholds for extreme event detection via spatial replicates as proposed by Mahecha et al. (2017) (for more details see Section 2.3).

The extreme event detection algorithm itself is applied to the set of hydrometeorological anomaly time series and returns anomaly scores computed by kernel density estimation. Kernel density estimation showed good performance among other possible methods and accounts for nonlinearities in the data (Flach et al., 2017). The resulting anomaly scores can be interpreted as a univariate index of deviation from the general multivariate pattern. We consider the highest 5% of the anomaly scores to be extreme events (95th percentile), which is within the typical range of percentiles defining extreme events (McPhillips et al., 2018). For a detailed step by step description to detect multivariate anomalies see Appendix A.

Note that the extreme events so far are multivariate extreme events in any direction of the variables, i.e. depending on the input variables. They may contain heatwaves as well as cold spells, droughts as well as extremely wet periods, as well as their compounding combinations. A selection of droughts and heatwaves takes place at a later step (see Section 2.4).

## 2.3 Climatic similarities to obtain spatial replicates

We follow the procedure described and developed by Mahecha et al. (2017), which was extended to the multivariate case by Flach et al. (2018). In summary, the used approach defines climatically and phenologically similar regions by using the leading principal components (here: three) of the seasonal cycles of the hydrometeorological variables (temperature, surface moisture, radiation) in addition to the vegetation proxy (gross primary productivity). Similar cycles appear in the same region of the obtained principal component space (Figure 1). Thus, a simple classification can be obtained by dividing the principal component space into equally sized cubes. Here we use 25 breaks for each of the first three principal components, which leads to 814 classes globally of similar climate and phenology. For each pixel, we sample four random spatial replicates from each region to efficiently run the anomaly detection workflow globally (previously the procedure was used for Europe only). The

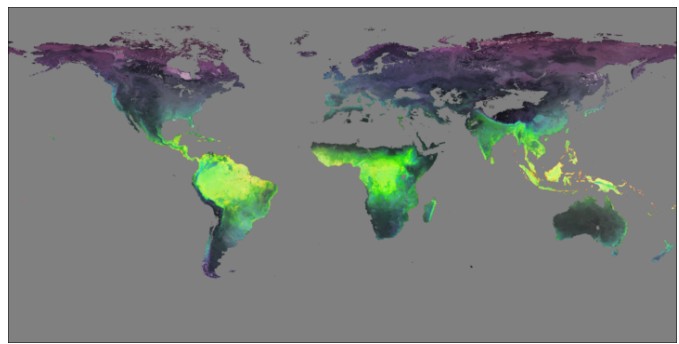

**Figure 1.** Map of the first three leading principal components (PCs) colored according to the colorspace hue (PC1), saturation (PC2), and, lightness (PC3). Coloring according to hue (i.e. the color wheel specifies which the tone of the color), saturation (intensity of the color ranging from grey to pure color) and lightness (brightness of the color, ranging from black to white) is one way to get an impression of three dimensional data on a map.

number of random spatial replicates depends on the number of observations in each 3 month period and the length of the time series (here: 16 years of data, each with 11 observations per 3 months period leads to 176 observations for each spatial replicate, thus 880 observations for the pixel and its 4 spatial replicates), which is a reasonable compromise between stability

of the results for extreme event detection and computational efficiency to run the anomaly detection procedure globally.

### 2.4 Framework for extracting event-based statistics

We use the extracted binary information (extreme / non-extreme) to compute statistics based on the spatio-temporal structure of the extreme events similar to (Lloyd-Hughes, 2011; Zscheischler et al., 2013; Mahecha et al., 2017; Chen et al., 2019). Extreme voxels are considered to belong to the same extreme event if they are connected within a 3 x 3 x 3 (long x lat x

time) cube. Note that this definition includes connections over edges. We compute event-based statistics from the 1000 largest extreme events globally as introduced also for the Russian heatwave (Flach et al., 2018). Specifically, we calculate affected volume, centroids, mean and integral of GPP separately for positive and negative anomalies, as well as the distance between the centroids of the positive and the negative anomalies of GPP during the event. We consider an event to be predominantly a relative drought (relative heatwave) if more than 50% of the surface moisture (temperature) values during the extreme event are

beneath (exceed) the 5th (95th) percentile of the variable. We select drought ($n = 98$) and heat ($n = 44$) events and combined drought–heat events ($n = 71$), which are taking place during the growing season (total $n = 213$). Growing season is defined here to be an extreme event taking place in the half year of the GPP maximum ($\pm$ 3 month). Our statistics account for the spherical geometry of the Earth by weighting with the cosine of latitude.

Furthermore, we evaluate if the positive and negative anomalies in GPP during the event predominantly have a spatial or

120 temporal component. Therefore, we split the event in parts with enhanced and parts with reduced productivity. Between those two parts, we compute the spatio-temporal distance between the centroids of each part. We consider positive and negative

(a) Agriculture

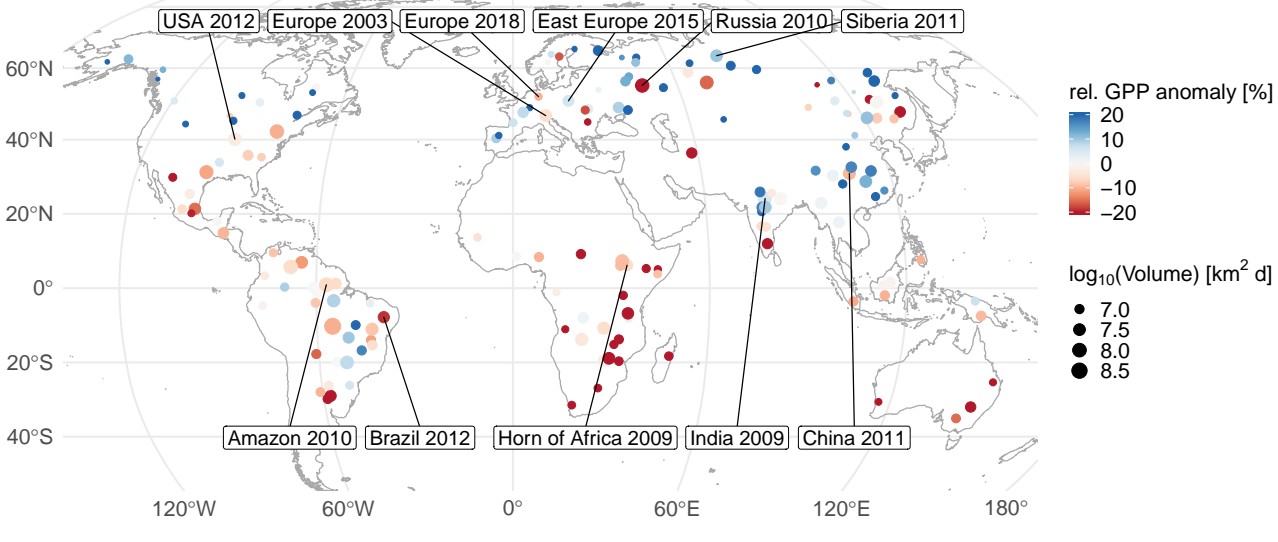

(b) Forests

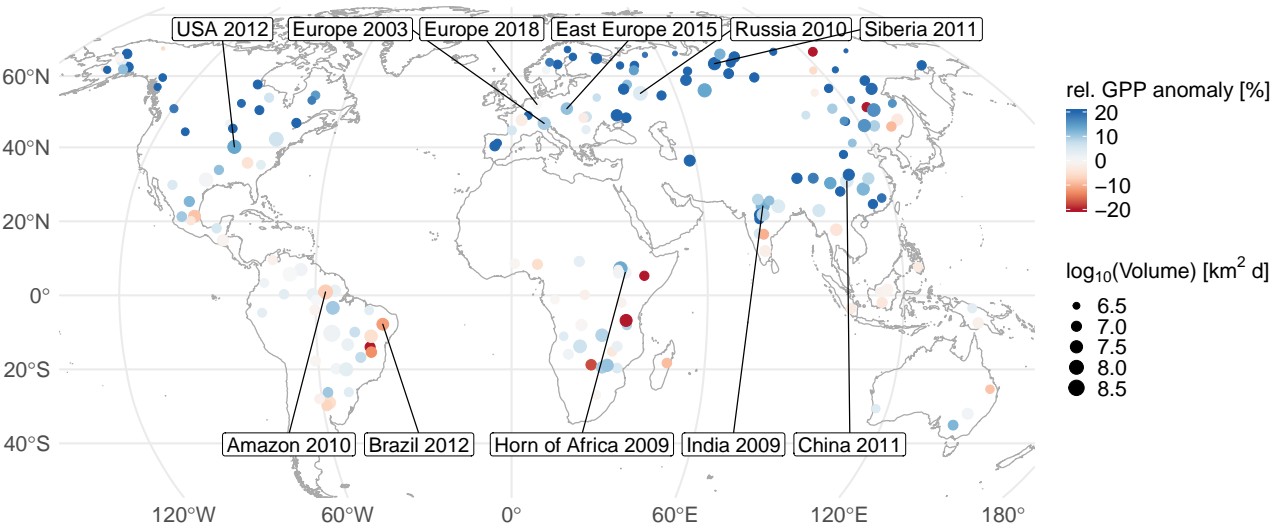

**Figure 2.** Relative drought and heat events colored with the relative anomaly in gross primary production. Figure is continued and described in detail on the next page.

(c) Other

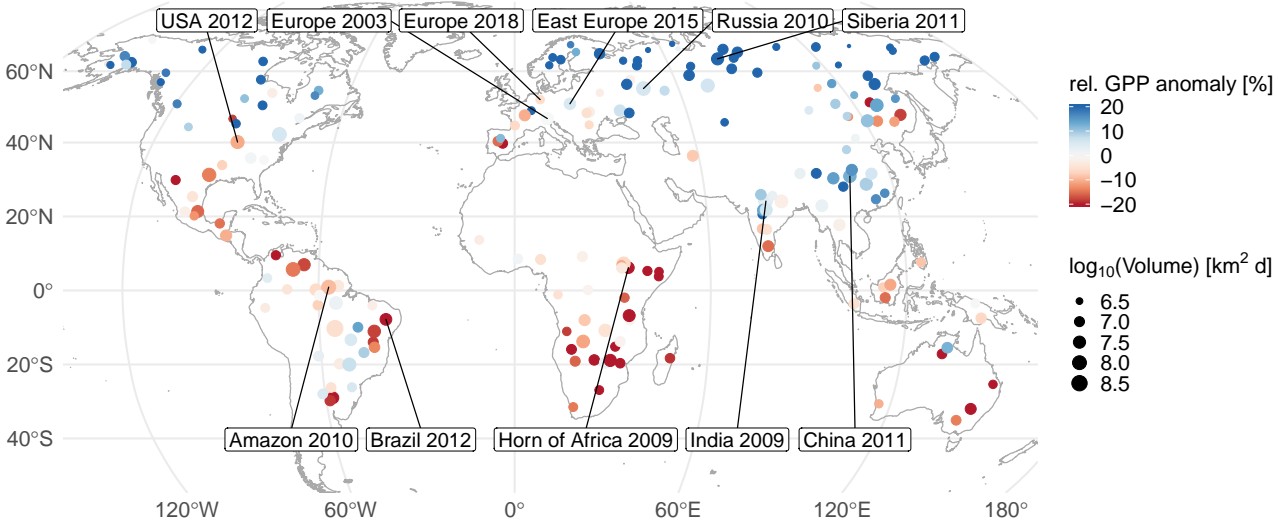

**Figure 2.** Relative drought and heat events colored with the relative anomaly in gross primary production for (a) agricultural, (b) forest and (c) other ecosystems (continued). Point sizes are proportional to the affected volume of the space-time event. The largest and some well known events are labeled. Note that one single extreme event can affect adjacent grid cells. Each of these adjacent grid cells may be dominated by a different ecosystem type. These extreme events will appear more than once, i.e. in (a), (b), and (c) each with the grid cells of part of the extreme event affecting the respective ecosystem. Labels are as follows: Compounding drought and heatwave in the United States 2012, most commonly known as US drought 2012 (USA 2012), compounding European drought and heatwave 2003, commonly known as European heatwave 2003 (Europe 2003), compounding European drought and heatwave 2018 (Europe 2018), compounding eastern European drought and heatwave 2015 (Europe 2015), Siberian heatwave 2011 (Siberia 2011), compounding western Russian drought and heatwave 2010, commonly known as Russian heatwave 2010 (Russia 2010), compounding Amazon drought and heatwave 2010, mostly known as Amazon drought 2010 (Amazon 2010), drought in Brazil 2012 (Brazil 2012), compounding drought and heatwave at the greater Horn of Africa 2009 (Horn of Africa 2009), compounding Indian drought and heatwave 2009 (India 2009), compounding drought and heatwave in China 2011 (China 2011).

.

GPP anomalies to occur predominantly spatially if the temporal distance of the centroids is almost simultaneous, i.e. less than one time step in the data (eight days). GPP anomalies are considered to be predominantly temporally changing if the spatial distance of the centroids is less than 110 km (approximately one degree at the equator). Both, spatial and temporal components 125  can be found for centroids which are more than 110 km and more than eight days away.

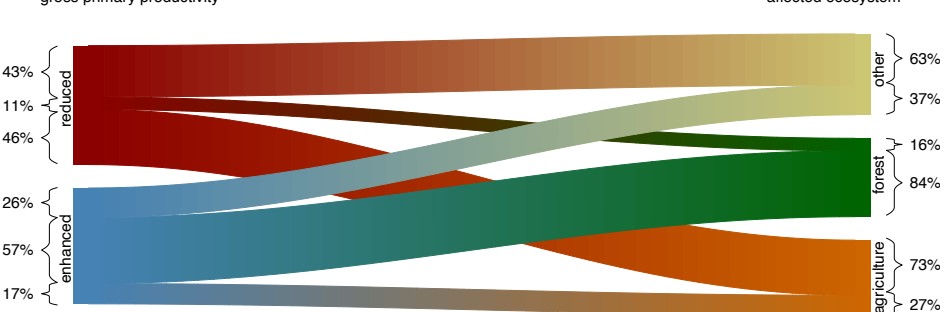

**Figure 3.** Proportion of GPP anomalies with reduced or enhanced productivity and their distribution in the different ecosystems (growing season events between 2003-2018). Bar sizes are proportional to the affected volume of the identified events. Numbers, denote percentages of the affected Volume for each of the categories. Forests tend to be associated with enhanced productivity rates, while agricultural ecosystems tend to be associated with reduced productivity.

## 2.5 Statistical model of GPP during extreme events

As we detect heatwaves and droughts relative to the mean seasonal patterns, positive or negative GPP anomalies during the droughts and heatwaves may additionally be influenced by differences in the conditions in the hydrometeorological variables during the extreme event, differences in background climate in which the vegetation is growing, or duration and affected area of the event. We use gradient boosting machines (Friedman, 2001) to predict average GPP anomalies during the event as a function of mean surface moisture, mean temperature, mean radiation during the event, duration, affected area, land cover class, and mean climate during the growing season, i.e. mean temperature and surface moisture during all growing seasons between 2003 and 2018. We tune model parameters by following a workflow described in Elith et al. (2008) using a hyper grid search from 100 different random initializations of splitting the data into training (75%) and testing (remaining 25%). We compute uncertainty of the variable importance measure described in (Friedman, 2001) from each of the 100 best models of the hyper grid search. Additionally we use an approach based on Local Interpretable Model-agnostic Explanations (LIME), which tries to predict each single observation in a black box model based on locally weighted regression (Ribeiro et al., 2016). Here, this approach helps to understand (1) the effect of specific land cover classes, and (2) the direction of the effect.

## 3 Results

The focus of this study is to better understand the impact of droughts and heatwaves on different vegetation types. Therefore we detect multivariate extreme events relative to the regional typical conditions during growing season. Furthermore, we use a global upscaled product of GPP to estimate the impact of the detected drought and heatwaves on different vegetation types. Our analysis based on a 5% threshold in the multivariate anomaly scores leads to a detection of 213 events (98 relative droughts, 44 relative heatwaves, 71 compound drought–heatwaves) between 2003 and 2018.

(a)

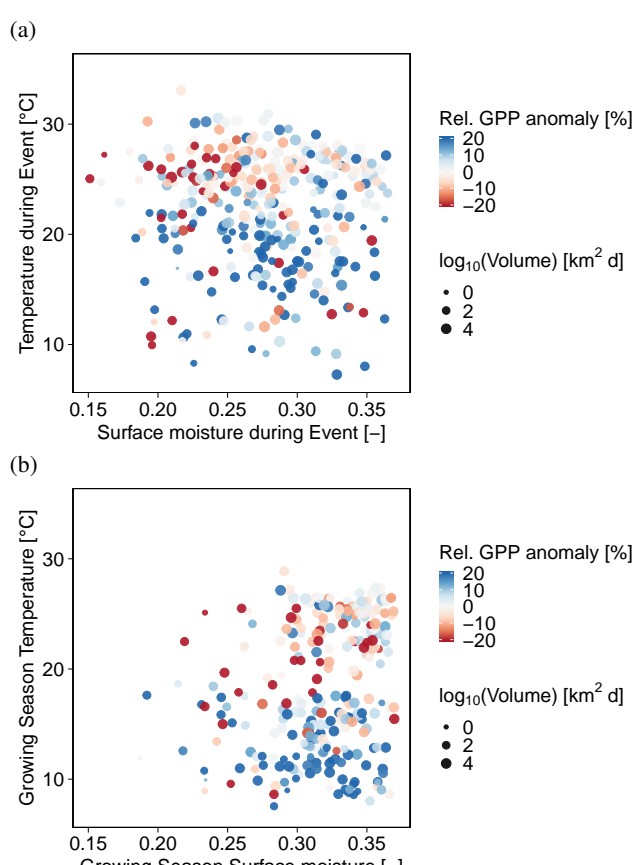

(b)

(c)

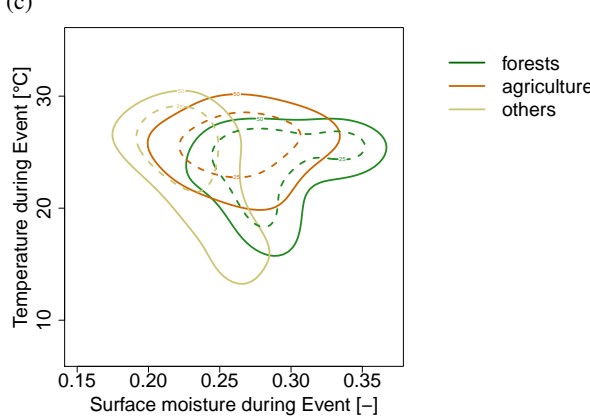

**Figure 4.** (a) Mean temperature and surface moisture during the relative drought and heat events and (b) growing season temperature and growing season surface moisture for forests and agricultural ecosystems. Size and color of the points denote the affected space–time volume and the direction of the impact on productivity. (c) Average conditions in temperature and surface moisture during the events for all ecosystems. Colored lines enclose 25% (stippled lines) and 50% (solid lines) of the events within forest, agricultural and other ecosystems.

If we only discriminate forest and agricultural ecosystems, we find substantial differences in the direction of the GPP anomalies during extreme droughts and heatwaves in the growing season. In agricultural (C3 and C4 croplands as well as C3 and C4 fractions croplands / natural vegetation mosaics) and other non-forest land-cover types (savannas, grasslands, open and closed shrublands, woddy savannas, permanent wetlands, urban and built-up), GPP was reduced during the identified events (agricultural land-cover types: 64% (56–72%, 95% confidence interval) reduction, Figure 2 (a); other ecosystems 60% (53–67%), Figure 2 (c)). In forested areas, instead, a majority of 71% (63-78%) of events shows enhanced productivity (Figure 2 (b)). The dichotomy described in the instantaneous response patterns confirms the overall statistics. Events with their centroid in France 2003, Russia 2010, and Germany 2018 all show bidirectional GPP anomalies that coincide with land-cover type transitions between predominantly forested land cover and others (a detailed illustration of the different events is provided in the supplementary materials). Figure 3 summarizes these findings across all events by relating the global integral areas of positive and negative anomalies in GPP during extreme events to the dominant land cover type. Note that the numbers in Figure 3 are proportions of the affected space-time volume of the extreme events and thus slightly different from the proportions of the number of events reported earlier in this paragraph. Thus, Figure 3 also indicates that it does not matter whether we obtain the statistics on an event basis or on the basis of a space-time volume. For both perspectives the main message is the same: agricultural and

other ecosystems are most strongly affected by droughts and heatwaves, whereas forests show neutral to enhanced productivity in the majority of the cases.

The events analyzed here are based on relative radiation, heat and water availability anomalies (see Methods). To better understand the role of absolute climate conditions we show the reported GPP anomalies in the terms of absolute temperatures and surface moisture levels in Figure 4(a). The figure shows that reduced rates of GPP tend to coincide with very low surface moisture and high temperature (eight-daily averages).

Furthermore, we show the events in climate space under which they occur, i.e. the average temperatures during growing season and average surface moisture during growing season (Figure 4(b)). Here, we can see that the events under scrutiny are detected as extreme events relative to the normal growing season conditions. Thus, the relative drought and heat events are occurring in very hot and dry climates (upper left of Figure 4(c)) as well as in very wet and cold climates (lower right of Figure 4(c)). We can see a tendency towards stronger negative impacts of heat and drought events in hotter climates (Figure 4(c)). A similar effect is not so clearly visible for very hot and dry climates. A reason may be a limited number of data points towards the upper left direction in Figure 4(c). Furthermore, heat and drought events in usually wet and cold climates are not associated with negative impacts or are even associated with an enhancement of productivity, e.g. when more radiation or temperature is available during the event in normally energy limited systems.

Delineating different ecosystems within this space shows that they are arranged along decreasing surface moisture values. Most extreme events in forests tend to occur under slightly higher surface moisture conditions compared to agricultural and other ecosystems (Figure 4(c)). Forests are hit less frequently critical dry conditions for which we predominantly observe reduced productivity. In contrast, we observe reduced productivity during the events for agricultural ecosystems, which experience frequently critical hot and dry conditions (Figure 4(c)).

Figure 4(a) shows that temperature and soil moisture have some effect on the direction of the impact, but does not consider other potentially important variables. Thus, we refine our understanding of the observed patterns using a statistical model. To unravel the importance of land cover type and other factors we predict average GPP anomalies using gradient boosting machines ($R^2 = 0.43$, Friedman (2001) Section 2.5) and explore their relative variable importance. Growing season temperature, event duration, land cover type, and surface moisture are, in decreasing order, the most important variables in the statistical model (Figure 5(a)).

Apart from identifying important variables that explain the GPP anomalies during drought and heat anomalies, we disentangle the direction of each factor's effect in the model, and, in particular for specific land cover classes. Negative model coefficients are a negative contribution of the respective variable to the GPP anomaly, i.e. the variable contributes to a stronger impact. In contrast, a positive model coefficient is associated with a positive contribution of the respective variable to the GPP anomaly. Thus, positive model coefficients weaken the impact of the extreme event, which may even lead to an enhancement of GPP during the extreme event.

Whereas growing season temperature and duration show a negative model coefficient, i.e. a longer duration and a warmer climate are associated with a stronger impact, a greater availability of radiation and higher surface moisture during the event reduce the impact on vegetation.

Productivity in different land cover types is influenced in contrasting ways: Forest ecosystems (Land cover types including 'forest' in its name) show increased average GPP during the extreme events. In contrast, agricultural ecosystems (land cover types including 'cropland' in its name) reduce average GPP anomalies (Figure 5(b)).

On land cover level, there is one exception of the agricultural ecosystems having a more neutral model coefficient. These are 'C3 croplands / natural vegetation mosaics'. However, 'C3 croplands' itself, 'C4 fraction croplands' and 'C4 fraction croplands / natural vegetation mosaics' all show negative coefficients. These agricultural systems are highly managed, so their difference may be more related to management than to ecological differences. Mostly in the temperate and boreal zone located mixed forests, deciduous broadleaf forests and evergreen needleleaf forests exhibit the most positive model coefficients. In the tropical zone located evergreen broadleaf forests show the least positive model coefficient. In between forests and grasslands and savannas, woody savannas have still considerably many trees in each grid cell. They are positioned with a positive to neutral model coefficient on the transition between forests and savannas. Savannas and grasslands are both associated with a negative model coefficient comparable to agricultural systems. Open and closed shrublands as well as permanent wetlands exhibit a negative coefficient. Urban and built-up is associated with a neutral coefficient.

We showed that the land cover type is one of the major factors influencing the direction of the GPP anomaly during an extreme event. A single hydrometeorological extreme event with a given magnitude and duration can affect two or more adjacent land cover types simultaneously with potentially contrasting impacts (spatial contrasting anomalies). Apart from an extreme event simultaneously affecting adjacent ecosystems with different or even contrasting impacts, it is also possible that one ecosystem shows contrasting impacts over time, i.e. with increasing duration. During startup of the extreme event enhanced productivity may be observed which can turn into a contrasting reduced productivity at a later stage of the extreme event. This temporal difference in the response with a longer lasting extreme event is considered to be a temporally contrasting anomaly. To explicitly quantify the role of spatial vs. temporal effects on the GPP anomalies during extreme events we split each event in parts with enhanced and reduced GPP anomalies and compute the centroidal distance in space and time. In fact, positive and negative GPP anomalies mostly co-occur simultaneously in adjacent spatial regions (116 events of 213 events in total within $\pm$ 8 days, Figure 6). Especially for large scale events (large volume), a considerable distance of the anomalies can be observed both in space as in time. Thus, these extreme events show spatial as well as temporal contrasting anomalies. Taking only the temporal distance into account, we have more events with enhanced productivity before reduced productivity (temporal distance $< -8$ days, $n = 44$) than events with reduced productivity before enhanced productivity ($> 8$ days, $n = 33$).

(a)

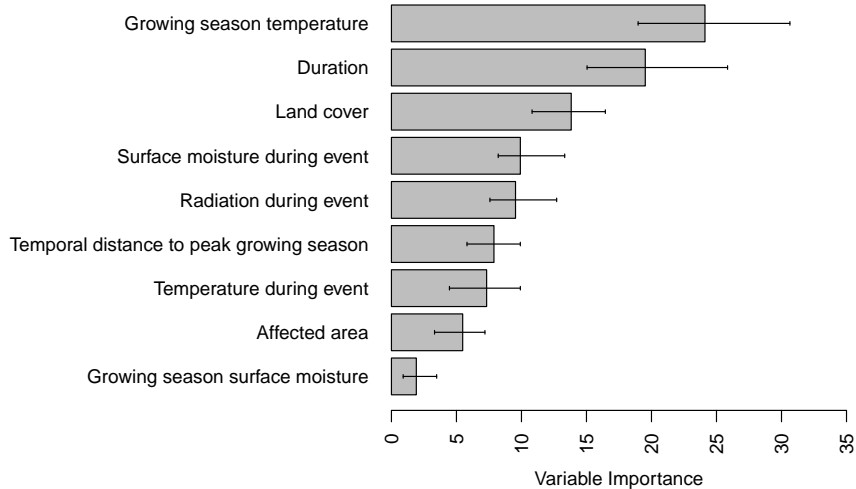

(b)

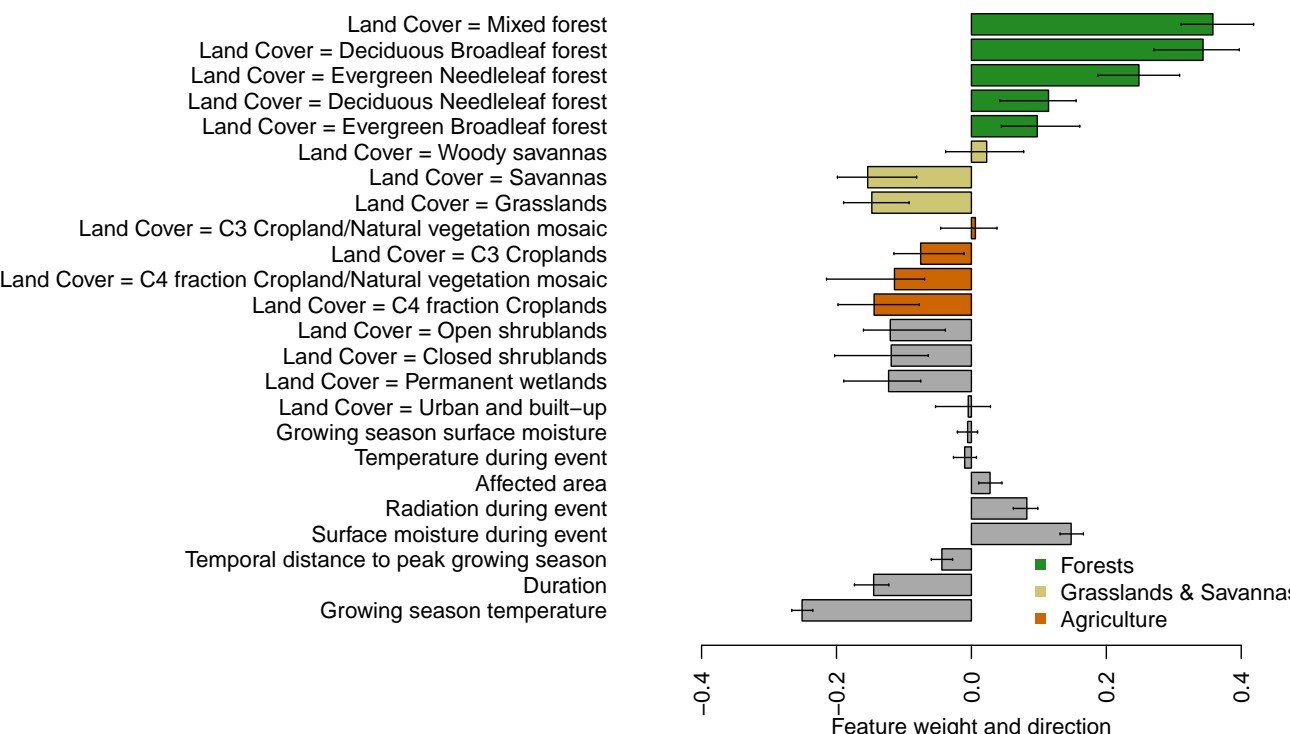

**Figure 5.** (a) Variable importance of the ten best gradient boosting machines predicting average GPP anomalies during the events, and (b) direction and feature weight of the variables explaining GPP anomalies of the individual events based on linear regression via local interpretable model-agnostic explanations (LIME).

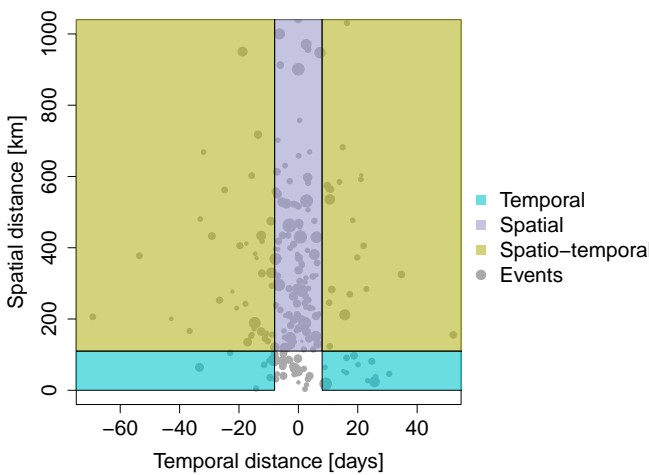

**Figure 6.** Each extreme event is split into parts with enhanced and reduced GPP anomalies. The centroidal distance between both parts in space and time shows whether contrasting GPP anomalies are predominantly taking place temporally, spatially or spatio-temporally. Point sizes are proportional to the event's affected volume.

## 4 Discussion

Contrasting responses of ecosystems to climate extremes, e.g. in the US in 2012 (Wolf et al., 2016) or in Russia in 2010 (Flach et al., 2018), are not singular cases but are shown to be frequent phenomena in response to hydrometeorological extreme events at the global scale. Within the same extreme event, reduced and enhanced productivity can be observed simultaneously in adjacent spatial regions. This finding complements previous studies on temporal (Wolf et al., 2016; Sippel et al., 2017a; Buermann et al., 2018) or spatial contrasting responses (Jolly et al., 2005; Zaitchik et al., 2006; Lewińska et al., 2016).

This study provides evidence that the impacts of extreme drought or heat anomalies on GPP during growing seasons are, firstly a function of event duration and long-term climate, but secondly, also depends on the affected land cover type. In particular the tendency towards positive vs. negative responses seems to be controlled by tree cover (similar to the results of Ivits et al. (2014); Walther et al. (2019)), i.e. forests seem to show higher resilience to drought and heat anomalies on the short term, which is reflected in a tendency towards positive GPP anomalies during the events. However, our results are based on events that are extreme relative to the regional normal conditions. In the supplementary materials we illustrate a range of events in more detail. For instance, a relative drought or heatwave in a typically wet ecosystem can boost productivity as well as a heatwave in ecosystems that are typically cold (see cases reported e.g. for China 20011, India 2009, and the Siberian heatwave 2011). Both water stress and temperature affect ecophysiological processes in a nonlinear manner. Heat events below optimal temperatures enhance photosynthesis (Wang et al., 2017), or photosynthesis may be enhanced by the radiation surplus during dry periods (Walther et al., 2019) especially at higher latitudes (Bachmair et al., 2018) and as long as ecophysiological limits are not violated. Yet, the prevalence of certain land cover types is partly controlled by climatic gradients, and therefore land cover cannot really be considered independently of the mean climatological conditions that likewise play a role (Figure 4(a)). Climate conditions also lead to adaptation of physiological processes. For instance, forests in dry ecosystems may be characterized by a more conservative water use strategy (Teuling et al., 2010; van Heerwaarden and Teuling, 2014; Ramos et al., 2015) and adapted to drought compared to analogous land cover types whose biogeographic history experienced colder and more moderate conditions (Doughty et al., 2015). Moreover, forests have access to deeper soil water compared to other ecosystems (Yang et al., 2016; Fan et al., 2017). The degree of isohydricity may further differentiate the response of forests, as it differs between tree species (Roman et al., 2015; Ruehr et al., 2015; Yi et al., 2017).

Our study only reports on GPP responses during the climatic anomaly without considering the legacy of the events. Responses may emerge with some time lag between weeks to months (Schwalm et al., 2012; Ruehr et al., 2015), or even at longer time scales (years) (Saatchi et al., 2013; Anderegg et al., 2015). Hence, finding enhanced productivity of forests during some heat event does not exclude increased mortality in the long-term. Forest ecosystems are known to potentially respond much delayed to environmental stress, which can trigger strong secondary impacts like insect outbreaks (Hicke et al., 2006; Rouault et al., 2006; Allen et al., 2010), or fires (Brando et al., 2014). In contrast, agricultural systems are known to be very directly vulnerable to droughts (De Keersmaecker et al., 2016; Bachmair et al., 2018). We choose the growing season as time period of interest, which is notably different than summer for some regions, e.g. in the Mediterranean where more positive responses

to warm anomalies in the cold season may be expected (Sippel et al., 2017b), and also impacts of droughts may be less than during the dry season (Huang et al., 2018).

Our results for gross primary productivity do not necessarily translate directly into net ecosystem exchange, because GPP and ecosystem respiration interact in a complex way (Richardson et al., 2007). However, studying the Russian Heatwave 2010 Bastos et al. (2014) found an increase in autotrophic respiration rates in forests, whereas crops declined their respiration rates. Flach et al. (2018) observed similar differences between forests and agricultural systems for gross primary productivity as well as for net ecosystem productivity during the Russian Heatwave. This similarity would suggest that the increase in autotrophic respiration for forest ecosystems during the heatwave does not offset potential carbon gains of available radiation and temperature in this particular energy limited forest ecosystem. Although, these findings remain case studies which are as such difficult to generalize, we would expect to see similar responses for net ecosystem productivity as for gross primary productivity.

Another aspect to discuss is data quality. We use ERA5 data for radiation and 2m-air-temperature. In particular for the latter one there are indications that 2m-air-temperature might be slightly underestimated: Land surface temperature is known to have a slight cold bias over the Iberian Peninsula due to effect of prescribed vegetation and topography (Johannsen et al., 2019). This bias might further translate into turbulent energy fluxes and eventually also affects 2m-air-temperature. However, as we use a relative detection scheme, a systematic seasonal cold bias in temperature would not change the occurrence of relative heat events in our study. In additions, it should be noted that ERA5 data has a considerably better data quality than its predecessor ERA-Interim (Johannsen et al., 2019), and is thus preferred in this study.

Furthermore, we use GLEAM surface moisture. GLEAM is driven by ERA5 data, thus errors in ERA5 might further propagate into GLEAM. Additionally, GLEAM is known to underestimate soil-moisture-temperature coupling due to soil and vegetation characteristics, in particular for temperate and continental climates (Gevaert et al., 2017). This may lead to an over-estimation of the remaining soil moisture in energy limited regimes and to an underestimation of soil moisture in water limited regimes. It implies an underestimation (overestimation) of drought intensity for energy (water) limited regimes in our study. However, GLEAM is still best in capturing latent heat flux dynamics compared to other products (Gevaert et al., 2017), and it therefore seems to be reasonable to rely on GLEAM to detect droughts and heatwaves in our study.

Gross primary productivity from FLUXCOM-RS may inherit errors from the underlying remote sensing products; these have, in particular, been discussed for tropical forests (Asner et al., 2004; Asner and Alencar, 2010; Wu et al., 2018). Recently, Stocker et al. (2019) showed at the global scale that remote sensing retrieved GPP underestimates drought impacts due to soil moisture effects on light use efficiency. Comparing our estimates of GPP impacts to published data from eddy covariance stations for two case studies (US 2012, (Wolf et al., 2016), and Europe 2003 (Ciais et al., 2005; Reichstein et al., 2007)) indicates that we do indeed underestimate GPP impacts. This lack of sensitivity of FLUXCOM-RS GPP to droughts and heatwaves seems to be a more general issue of GPP estimates as well as in remote sensing in general: we suspect that in addition to the GPP estimates used by Stocker et al. (2019), also FLUXCOM-RS GPP underestimates the impacts of climate extreme events specifically for forest ecosystems. FLUXCOM-RS additionally exhibits a good agreement for forests globally

with GPP estimates based on solar-induced fluorescence (Walther et al., 2019). Thus, the lack of sensitivity to drought and heat impacts in forest ecosystems may be a more general issue in remote sensing data.

## 5 Conclusions

To understand the effect of different vegetation types and other factors to the response of drought and heatwaves we analyzed 213 events between 2003 and 2018 globally. Generally, we find that extreme events of a given extent, magnitude and duration often affect different adjacent vegetation types, each vegetation type differing in their specific response to the event. Quantifying these findings, we find that vegetation is one important variable which has to be considered for understanding the impact of climate extremes. Whereas agricultural systems, grasslands, savannas and shrublands are most impacted in terms of gross primary productivity, forests are not particularly sensitive to the extreme event or even show enhanced gross primary productivity during the events.

Thus, we conclude that a more differentiated consideration of the role of land cover reveals firstly major differences between forests, agricultural and other ecosystems. These differences may originate from a different (micro-)climate or different water management strategies including the access to deeper soil water or point to more strongly lagged impacts in forest ecosystems.

Our findings imply for future climate that forest ecosystems may be crucial for mitigating immediate negative impacts on the carbon cycle of an increasing number of heatwaves. However, longer lasting heatwaves, drying in continental climates or a disproportionate increase in summer drought–heat events due to mutual dependencies may lead more frequently to critical moisture conditions for which we observe negative impacts for forests and to which forests are not well adapted to. This is particularly critical as forest recovery times are multi-decadal.

However, the lack of sensitivity of forest ecosystems to droughts and heatwaves is stronger than we would expect it to be, as forests are generally considered to be vulnerable to drought and heat related mortality risks. Thus, we think that our results also point towards deficiencies in FLUXCOM-RS derived GPP which are potentially a more general issue in remote sensing derived indices of vegetation activity. These deficiencies call for the development of new global GPP products with a higher sensitivity to droughts and heatwaves, which can unravel the role of forest ecosystems in a more frequently hot and dry future climate.

*Data availability.* We use data originating from the FLUXCOM initiative (http://www.fluxcom.org), the GLEAM model data integration framework (https://www.gleam.eu/), and ERA5 (https://cds.climate.copernicus.eu/cdsapp#!/home). The harmonized data set is available within the project Earth System Data Lab (ESDL) and can be accessed here: https://www.earthsystemdatalab.net/index.php/interact/data-lab/.

## Appendix A: Details on the procedure to detect anomalies

The procedure which is used here to detect multivariate anomalies works as follows (see also Flach et al. (2018)):

1. select one pixel and some spatial replicates (here: four spatial replicates as defined by Section 2.3) to obtain five considerably similar time series of temperature, radiation and surface moisture.

2. subtract a smoothed median seasonal cycle from each time series to obtain anomalies (deviations from the normal seasonality) and their covariance matrix Q.

3. select a seasonal window of 3 months in each years (three months would correspond to e.g. all summers in the years under scrutiny).

4. standardize the anomalies to zero mean and unit variance.

5. compute kernel density estimates using a standard multivariate normal kernel K with the covariance matrix Q. Using a multivariate normal kernel accounts for linear correlations among the set of input variables (here: radiation, temperature, surface moisture), while allowing for non-linear shapes of the data (Flach et al., 2017).

6. transform the resulting univariate index of deviations from the general multivariate pattern into a score of normalized ranks between 0.0 (very normal) and 1.0 (extremely far away from the dense regions of the multivariate distribution)

7. select the data points higher than a threshold of 0.95 to obtain 5% of the data as multivariate extreme events.

8. memorize the extreme events and the obtained score for the selected pixel and season

9. repeat the procedure (3-8) in a running moving window of 3 months length

10. repeat the procedure with the next pixel.

*Author contributions.* MF and MDM designed the study in collaboration with AB, FG, SS, MR. MF conducted the analysis and wrote the manuscript with contributions from all co-authors.

*Competing interests.* The authors declare they have no conflict of interests.

*Acknowledgements.* This research was supported by the European Space Agency (project "Earth System Data Lab") and the European Union's Horizon 2020 research and innovation programme (project "BACI", grant agreement no 64176). The authors are grateful to the FLUXCOM initiative (http://www.fluxcom.org) for providing the data. MF acknowledges support by the International Max Planck Research School for Global Biogeochemical Cycles (IMPRS). Two reviewers provided valuable feedback for improvement.

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
