# Peer review of "Vegetation modulates the impact of climate extremes on gross primary production"

_Biogeosciences, 2020_

## Referee Comment (RC1) · Anonymous Referee #1 · 15 Apr 2020

The study deals with the role of vegetation for the effects of climate extremes on gross primary production (GPP). This is by analysing a selection of different observational data sets for the last 15 years or so. Although I find the subject of the study interesting and highly relevant, I don't find its presentation in the manuscript meets the quality standard, making it suitable for publication in its current form. Therefore, in my opinion, the manuscript should undergo a major revision before being published in Biogeoscienes. I will further explain my reservations in the following:

General comments:

1. In the study, forests are combined over the whole globe, providing estimates of the impacts of droughts and/or heatwaves on GPP at a global scale. I wonder, whether it would add value to the study, if also different categories of trees or different climate

ranges (which typically also have a dominating type of trees) were distinguished. Different types of trees in a different background climate might be affected by the extreme events in different ways.

2. I find that the presentation of the results (Section 3) only makes up a relatively small part of the paper, certainly as compared to the introduction and the section on the methodology. I think this section needs to be extended to have a more balanced paper.

3. I find that the conclusions (Section 5) of the paper a bit weak. I think they could be extended in several ways, e.g. what the findings of the study mean for the terrestrial carbon budget and carbon dioxide concentrations under climate change.

4. I am a bit confused that some of the dots in Fig. 1 seem to be assigned to different types of ecosystems. Unless this is related to the way of presentation, it needs to be explained that grid points can comprehend different types of ecosystems and that in the analysis all (my assumption) types of ecosystems are included rather than the dominating type. I also wonder, whether, if in fact different types are considered, there should be a lower limit on the extent/fraction of the area covered by each type in a grid point.

5. I miss information on the types of ecosystem that are considered in the study in various places. Actually, it seems the only place, where this information can be obtained, is in Fig 4b. The information could easily be provided in a table in Section 2, where the ecosystems could also be grouped in the three main categories: forest, agriculture and others.

6. I miss a discussion of the limitations and potential biases of the data used in the study. This is only done for the FLUXNET data in the discussion (Section 4).

Specific comments:

Abstract

7. Page 1, lines 10-11: "On the other hand... droughts and heatwaves." – That would

actually mean a limitation of the data, which to my understanding hasn't been discussed in the paper.

Introduction

8. General: I would find a short paragraph on the structure of the paper at the end of the introduction really helpful.

9. Page 2, line 27: "the crucial role of timing" – I assume this refers to the timing of the extreme events. Please clarify.

10. Page 2 line 31: "the least understood aspect" – I wonder whether there is a review paper on this or another suitable reference to support this statement.

11. Page 2, line 39: "in some meteorological... in ecological processes" – I am not sure, what this statement means. Please clarify.

12. Page 2, lines 95-96: "extreme relative to their expected value" – I am not sure that I understand this. In any case, considering a global absolute threshold would not make much sense, while it would make sense to use locally varying thresholds based on the same percentile, e.g. the 95th percentile, would.

Method

13. General: I think it would be nice to properly introduce the acronyms of the various datasets.

14. Page 3, line 55: "ERA5" – I think it need to be mentioned that in ERA5 vegetation doesn't vary but is prescribed via some climatological value. That has an effect on the turbulent energy fluxes at the land surface and, thus, might also affect the near-surface temperature.

15. Page 3, line 57: "GLEAM model-data integration framework" – It would be interesting to know how and to which extent these data are constrained by observations.

16. Page 3: line 62: "2003-2018 period" – The choice of this particular time period for the study is not motivated at all.

17. Page 3, line 71: "for more details see the B" – It is not clear, what this means and what it refers to. Appendix B, maybe (see also my comment below)?

Results

18. Page 6, line 115: "non-forested land-cover types" – This is one of the (many) places, where information on the types of ecosystems is missing. See also my comment above.

19. Page 8, lines 136-137: "the most important... model" – I find it interesting to note that according to this statistical model soil moisture doesn't seem to play a role. This is, however, in contrast to the results presented in Fig. 4b, where soil moisture receives a rather large weight. I wonder, how these – at first sight – contrasting results can be reconciled.

20. Page 8, lines 148-149: "but enhanced productivity... contrasting anomalies)" – I am not sure what this statement means. Please clarify.

Discussion

21. General: I think it would be important to also discuss the potential implications of the effects of extremes on net ecosystem productivity (NEP), given the effects on GPP, to the extent possible.

Conclusions

22. General: I think the conclusions need to fill more than the one short paragraph (see my comment above). I also wonder, whether it would be helpful with a short summary of the main results of the study.

Appendix

23. General: I find the appendix unnecessary. This is because, in my view, Fig. A1 should be part of the section on the results (it is discussed quite a bit and is needed to give a complete picture) and Fig. B1 doesn't provide much relevant information (and is not really referred to).

Figures

24. Figure 1: One of the prominent extreme events ("Russia 2010") is not linked to a dot in the figure. Is this a mistake or doesn't exist a particular grid point that can be assigned to this event? Also, I think this figure should be extended with the panel representing "other ecosystems", now Fig. A1 in the appendix.

Supplementary material

25. General: An introduction into the structure of the figures, i.e. what the different panels show and how they relate to each other. Also, I think it would be helpful to give the "identification" of the extreme period and the type of extreme (drought, heat wave or a compound even) in a headline. I understand the rational for presenting mean values for temperature and soil moisture, but presenting anomalies instead might highlight some of the regional details and would indicate the soil moisture/temperature coupling. Also, an indication of the colours/numbers of the different ecosystem types shown in the figures would be helpful. That could also be part of the introduction to the supplementary material. See also my comment above.

---

## Referee Comment (RC2) · Anonymous Referee #2 · 17 Apr 2020

The paper investigates the importance of land cover type in controlling the impacts of climate extremes relative to other factors using a global upscaled product of GPP. The results show that heat and drought events seem to reduce GPP in grasslands and agricultural areas and to increase GPP in forests. The work calls for considering different land cover types in the assessments of the impact of climate extremes on ecosystem functioning. Overall, the objectives of the paper are clear. However, some methodology and results still need further improvement, and some Figure needs to do some improved. I would recommend a major revision. Detailed comments are listed below: 1. Figure 1 is not intuitive enough; it needs some improvement. It should label the specific events name rather than region and year. 2. I suggest Figure 2 need to label the proportion value. 3. Figure 3a is too orderless. I suggest it needs not to label the

events. 4. The authors group land cover classes in forest and agricultural ecosystems, what about grasslands? Abstract illustrates GPP in grasslands is generally reduced during heat and drought events. And which year the land cover data is? 5. I am so fusing about the methodology; I suggest to introduce more detailed of the method about preprocessing and anomaly detection. 6. The results section needs further analysis, especially need quantitative analysis.

---

## Author Comment (AC1) · 30 May 2020

*Response on the*
Reviewer: The study deals with the role of vegetation for the effects of climate extremes on gross primary production (GPP). This is by analysing a selection of different observational data sets for the last 15 years or so. Although I find the subject of the study interesting and highly relevant, I don't find its presentation in the manuscript meets the quality standard, making it suitable for publication in its current form. Therefore, in my opinion, the manuscript should undergo a major revision before being published in

[Figure]

Biogeoscienes. I will further explain my reservations in the following:

*Response: We are pleased that the reviewer is considering our manuscript highly relevant and are confident that we are able to present the manuscript in a form meeting the reviewers expectations as well as the quality standard of the journal. Please find the responses to the individual comments below.*

**General comments:**

Reviewer: 1. In the study, forests are combined over the whole globe, providing estimates of the impacts of droughts and/or heatwaves on GPP at a global scale. I wonder, whether it would add value to the study, if also different categories of trees or different climate ranges (which typically also have a dominating type of trees) were distinguished. Different types of trees in a different background climate might be affected by the extreme events in different ways.

*Response: We would like to thank the reviewer for this comment. Differente climate ranges are distinguished in this study so far by using growing season temperature and growing season surface moisture as drivers of the statistical models explaining the impacts of the extreme events. Figure 3 shows the impact of the extreme events using temperature and surface moisture during the event. To further distinguish these impacts in different climate ranges, we will add a similar second plot showing the extreme events in climate space opened up by growing season temperature and growing season surface moisture. We will add a paragraph on different climate ranges to the result section. Regarding the second aspect, distinguishing more different categories of trees, we would like to note here, that although forests are combined over the whole globe, the ecosystem type forests provides an astonishing homogeneous response pattern globally. We further differentiated forests into their different land cover classes (such as evergreen needleleaf forest, mixed forest, . . .) in Figure 4b and will provide more details on the specific land cover classes in the result section. However, further splitting different tree categories up to a species level is not possible so far to the best*

*of our knowledge by the means of globally available remote sensing products as used in this study.*

Reviewer: 2. I find that the presentation of the results (Section 3) only makes up a relatively small part of the paper, certainly as compared to the introduction and the section on the methodology. I think this section needs to be extended to have a more balanced paper.

*Response: We will extend the presentation of the results in a revised version of the paper. Therefore, we will add a second figure similar to Figure 3 and a paragraph on the climate space. Furthermore, we will add more details on specific land cover types in the results section (see also 18.) and provide more details on our findings on spatial vs. temporal contrasting anomalies (see also 20.).*

Reviewer: 3. I find that the conclusions (Section 5) of the paper a bit weak. I think they could be extended in several ways, e.g. what the findings of the study mean for the terrestrial carbon budget and carbon dioxide concentrations under climate change.

*Response: We would like to thank the reviewer for the suggestions on how to improve the conclusions (see also 22.). We will add a summarizing paragraph of the main results at the beginning of the conclusion and we will add a paragraph of our findings with respect to climate change and the carbon cycle (on p.12, l. 204): "[...] lagged impacts in forest ecosystems. Our findings imply for future climate that forest ecosystems may be crucial for mitigating immediate negative impacts on the carbon cycle of an increasing number of heatwaves (Seneviratne et al. 2012, Coumou & Robinson 2013). However, longer lasting heatwaves, drying in continental climates (Meehl et al. 2000) or a disproportionate increase in summer drought–heat events due to mutual dependencies (Zscheischler & Seneviratne 2017) may lead more frequently to critical moisture conditions for which we observe negative impacts for forests and to which forests are not well adapted to (Isaac-Renton et al. 2018). This is particularly critical as forest recovery times are multi-decadal. However, the lack [...]"*
Reviewer: 4. I am a bit confused that some of the dots in Fig. 1 seem to be assigned to different types of ecosystems. Unless this is related to the way of presentation, it needs to be explained that grid points can comprehend different types of ecosystems and that in the analysis all (my assumption) types of ecosystems are included rather than the dominating type. I also wonder, whether, if in fact different types are considered, there should be a lower limit on the extent/fraction of the area covered by each type in a grid point.

*Response: We would like to thank the reviewer for pointing us to this possible source of misunderstanding. We will extend the explanation of Fig. 1: it encompassed all grid cells affected by the extreme event. In many cases one extreme event affects adjacent grid cells which may each be dominated by a different ecosystem type. However, each grid cell (at the resolution of 1/12 degree) has still one dominating ecosystem type.*

Reviewer: 5. I miss information on the types of ecosystem that are considered in the study in various places. Actually, it seems the only place, where this information can be obtained, is in Fig 4b. The information could easily be provided in a table in Section 2, where the ecosystems could also be grouped in the three main categories: forest, agriculture and others.

*Response: We will provide the requested table in section 2 (methods) and will provide more information on the different types of ecosystems (see 18.).*

Reviewer: 6. I miss a discussion of the limitations and potential biases of the data used in the study. This is only done for the FLUXNET data in the discussion (Section 4).

*Response: We thank the reviewer for this important note. Indeed, we specifically discussed the limitations and potential biases of FLUXCOM-RS data as we consider this for the findings of our study to be particularly important. We will add a section to the Discussions on the limitations of temperature and radiation (from ERA5) as well as surface moisture (from GLEAM).*

**Specific comments:**

Abstract

Reviewer: 7. Page 1, lines 10-11: "On the other hand. . . droughts and heatwaves." – That would actually mean a limitation of the data, which to my understanding hasn't been discussed in the paper.

*Response: We would like to thank the reviewer for this comment. We would like to note that the limitations of FLUXCOM-RS are discussed as already mentioned earlier (6., see above) on p. 12, l.191-200. However, the discussion does not explicitly mention the lack of sensitivity to droughts and heatwaves. We will extend the discussion to explicitly mention the lack of sensitivity to droughts and heatwaves and we will add a paragraph discussing the limitations of temperature, radiation and surface moisture data as outlined above (6.).*

Introduction

Reviewer: 8. General: I would find a short paragraph on the structure of the paper at the end of the introduction really helpful.

*Response: We will add a short paragraph on the structure of the paper at the end of the introduction as suggested.*

Reviewer: 9. Page 2, line 27: "the crucial role of timing" – I assume this refers to the timing of the extreme events. Please clarify.

*Response: Yes, indeed. We will clarify it to be "crucial role of timing of the extreme event"*

Reviewer: 10. Page 2 line 31: "the least understood aspect" – I wonder whether there is a review paper on this or another suitable reference to support this statement.

*Response: We apologise for this statement being a bit speculative and change it into "one important aspect".*

Reviewer: 11. Page 2, line 39: "in some meteorological. . . in ecological processes" –
I am not sure, what this statement means. Please clarify.

*Response: We clarify it as follows: "One option is to use values over some global
thresholds to detect extremes e.g. to detect temperatures above 25 or 30 degree
Celsius and to investigate the associated anomaly in vegetation productivity.*

Reviewer: 12. Page 2, lines 95-96: "extreme relative to their expected value" – I am not
sure that I understand this. In any case, considering a global absolute threshold would
not make much sense, while it would make sense to use locally varying thresholds
based on the same percentile, e.g. the 95th percentile, would.

*Response: We fully agree with the reviewer. We will change it into: "Another option
is to define extreme events relative to some locally varying threshold, e.g. defined by
the 95th percentile of the distribution of the data. Here, we rely on the latter definition,
and refine the definition by taking also a joint multivariate distribution of the data with
regionally varying thresholds into account."*

Method

Reviewer: 13. General: I think it would be nice to properly introduce the acronyms of
the various datasets.

*Response: Yes, we will properly introduce the acronyms of the data sets as requested.*

Reviewer: 14. Page 3, line 55: "ERA5" – I think it need to be mentioned that in ERA5
vegetation doesn't vary but is prescribed via some climatological value. That has an
effect on the turbulent energy fluxes at the land surface and, thus, might also affect the
near-surface temperature.

*Response: We would like to thank the reviewer for this important comment and will
mention it in the extended discussion of the data limitations.*

Reviewer: 15. Page 3, line 57: "GLEAM model-data integration framework" – It would

be interesting to know how and to which extent these data are constrained by observations.

*Response: This is indeed an important aspect. GLEAM is driven by precipitation and microwave satellite observations to estimate soil moisture. Surface net radiation and near surface air temperature are used to estimate evaporation.*

Reviewer: 16. Page 3: line 62: "2003-2018 period" – The choice of this particular time period for the study is not motivated at all.

*Response: This choice represents the common time period of all data sets used. It is mainly constrained by GLEAM v3.3.b (starting 2003, ending 2018) and FLUXCOM-RS (starting 2001, ending 2018). We will add the following sentence: "The time period is chosen as it represents the common period of all data sets used at the time of the analysis."*

Reviewer: 17. Page 3, line 71: "for more details see the B" – It is not clear, what this means and what it refers to. Appendix B, maybe (see also my comment below)?

*Response: We will add the word appendix, which we unfortunately missed in the current version of the manuscript.*

Results

Reviewer: 18. Page 6, line 115: "non-forested land-cover types" – This is one of the (many) places, where information on the types of ecosystems is missing. See also my comment above.

*Response: We will specify the non-forested land-cover types ("savannas, grasslands, open and closed shrublands, permanent wetlands") as well as the agriculture land cover type ("C3 and C4 croplands as well as C3 and C4 fractions croplands / natural vegetation mosaics"). Furthermore, we will extend the description of Figure 4(b) on p.8 l. 140-144 with more details of the results with respect to the different land cover types.*

Reviewer: 19. Page 8, lines 136-137: "the most important. . . model" – I find it interesting to note that according to this statistical model soil moisture doesn't seem to play a role. This is, however, in contrast to the results presented in Fig. 4b, where soil moisture receives a rather large weight. I wonder, how these – at first sight – contrasting results can be reconciled.

*Response: We thank the reviewer to point us to this important aspect. We apologise that this aspect can be misunderstood. We do not state nor do we want to state that soil moisture does not play a role. Soil moisture is one important variable in the statistical model, which we definitely should mention. We will mention that surface moisture is the fourth most important variable after land cover type, as can be seen from Fig. 4(a) in line 137. Furthermore, we will tone down the first sentence of the paragraph to "Figure 3(a) shows that temperature and soil moisture have some effect on the direction of the impact, but does not consider other potentially important variables. Thus, we refine our understanding of the observed patterns using a statistical model."*

Reviewer: 20. Page 8, lines 148-149: "but enhanced productivity. . . contrasting anomalies)" – I am not sure what this statement means. Please clarify.

*Response: We will reformulate and extend the statement as follows: ". . . (spatial contrasting anomalies). Apart from an extreme event simultaneously affecting adjacent ecosystems with different or even contrasting impacts, it is also possible that one ecosystem shows contrasting impacts over time. During startup of the extreme event enhanced productivity may be observed which can turn into a contrasting reduced productivity at a later stage of the extreme event. This temporal difference in the response with a longer lasting extreme event is considered to be a temporally contrasting anomaly. To explicitly quantify . . . "*

Discussion

Reviewer: 21. General: I think it would be important to also discuss the potential implications of the effects of extremes on net ecosystem productivity (NEP), given the

effects on GPP, to the extent possible.

*Response: We will add a sentence on the implications for net ecosystem exchange on p. 12, l 190.*

Conclusions

Reviewer: 22. General: I think the conclusions need to fill more than the one short paragraph (see my comment above). I also wonder, whether it would be helpful with a short summary of the main results of the study.

*Response: We will add a paragraph with a short summary of the main results to the conclusions and will add a sentence on our findings with respect to climate change and the carbon cycles as stated above (3.)*

Appendix

Reviewer: 23. General: I find the appendix unnecessary. This is because, in my view, Fig. A1 should be part of the section on the results (it is discussed quite a bit and is needed to give a complete picture) and Fig. B1 doesn't provide much relevant information (and is not really referred to).

*Response: We will include Figure A1 into the result section as requested. However, we would like to leave Figure B1 in the Appendix, as it illustrates the found regions with similar seasonal cycles which are used to obtain similar thresholds in the multivariate extreme event detection procedure.*

Figures

Reviewer: 24. Figure 1: One of the prominent extreme events ("Russia 2010") is not linked to a dot in the figure. Is this a mistake or doesn't exist a particular grid point that can be assigned to this event?

*Response: We apologise that the linking line of Russia 2020 is hidden behind "Siberia 2011" at the very beginning. We will ensure that the link is visible in a revised version*

*of the manuscript.*

Also, I think this figure should be extended with the panel representing "other ecosystems", now Fig. A1 in the appendix.

*Response: We will move Figure A1 ("other ecosystems") to the results section.*

Supplementary material

Reviewer: 25. General: An introduction into the structure of the figures, i.e. what the different panels show and how they relate to each other. Also, I think it would be helpful to give the "identification" of the extreme period and the type of extreme (drought, heat wave or a compound even) in a headline. I understand the rational for presenting mean values for temperature and soil moisture, but presenting anomalies instead might highlight some of the regional details and would indicate the soil moisture/temperature coupling. Also, an indication of the colours/numbers of the different ecosystem types shown in the figures would be helpful. That could also be part of the introduction to the supplementary material. See also my comment above.

*Response: We will revise the Supplementary material. Specifically, we will add a general introduction for the structure of the figures and we will add the type of the extreme in a headline. However, we would like to present the figures with mean values as they currently are. The rationale behind presenting mean values instead of relative anomalies is to illustrate the range of global temperatures and surface moisture during extreme events (which are already detected by a relative approach).*

---

## Author Comment (AC2) · 30 May 2020

*Response on* the interactive comment by Anonymous Referee 2

Reviewer: The paper investigates the importance of land cover type in controlling the impacts of climate extremes relative to other factors using a global upscaled product of GPP. The results show that heat and drought events seem to reduce GPP in grasslands and agricultural areas and to increase GPP in forests. The work calls for considering different land cover types in the assessments of the impact of climate extremes on ecosystem functioning. Overall, the objectives of the paper are clear. However, some methodology and results still need further improvement, and some Figure needs to do

some improved. I would recommend a major revision. Detailed comments are listed below:

*Response: We would like to thank the author for the feedback on our manuscript. We address the comments in the following more specifically.*

Reviewer: 1. Figure 1 is not intuitive enough; it needs some improvement. It should label the specific events name rather than region and year.

*Response: We would like to thank the reviewer for this comment. As the space in the figure itself is limited, we would prefer to add specific names rather to the caption of the figure, than to the figure itself. However, we would like to note that some of the events have a well known name (e.g. Russian Heatwave 2010, Amazon Drought 2010, European Heatwave 2003, . . .) but some do not have well known or clearly defined names (e.g. Siberia 2011, Horn of Africa 2009).*

Reviewer: 2. I suggest Figure 2 need to label the proportion value.

*Response: We fully agree with the reviewer and would like to thank the reviewer for this suggestion. We will provide labels for the proportions in a revised version of the manuscript.*

Reviewer: 3. Figure 3a is too orderless. I suggest it needs not to label the events.

*Response: We will remove the event specific labels from the figure as requested.*

Reviewer: 4. The authors group land cover classes in forest and agricultural ecosystems, what about grasslands? Abstract illustrates GPP in grasslands is generally reduced during heat and drought events. And which year the land cover data is?

*Response: We will add more details about grasslands to the result section. Figure 4(a) shows that they have a general negative response coefficient in the impact model. Of course, we will add the information of the land cover data to a revised version of the manuscript.*

[Figure]

Reviewer: 5. I am so fusing about the methodology; I suggest to introduce more detailed of the method about preprocessing and anomaly detection.

*Response: We will add more details about preprocessing and anomaly detection in the method section as requested.*

Reviewer: 6. The results section needs further analysis, especially need quantitative analysis.

*Response: We would like to note that the result section provides quantitative statistics on which our findings are based. For instance, we provide fractions of the events with reduced / enhanced productivity including estimates of uncertainty, or we identify the main drivers of the ecosystems response based on gradient boosting machines. We would be very pleased if the reviewer could provide more details on which quantitative analysis he is specifically aiming for.*

---

## Editor Comment (EC1) · Bart van den Hurk (Editor) · 2 Jun 2020

The two reviewers are both positive about the relevance and level of interest of the analysis, and generally subscribe the followed approach and data used. Both reviewers also make suggestions for further analysis and clarifications, and agree on a desire to elaborate more on the results and implications of the study.

The authors have given a detailed point-by-point reply to the review comments, and agree with nearly all comments made. Also they make solid proposals for accommodating these comments, thereby making proper use of the review process to clarify the manuscript and increase the level of interest even further.

I do agree with the reviewers assessments and also appreciate the authors' willingness

and ability to modify their manuscript accordingly. I do invite the authors to follow this approach, and will check on the degree to which the authors' intentions to reply to the reviews are being met in a new version of the manuscript. If this revision is in my view succesfull in addressing the reviewers' needs I will not send it out for another review round

---

## Author Response (AR1)

The study deals with the role of vegetation for the effects of climate extremes on gross primary production (GPP). This is by analysing a selection of different observational data sets for the last 15 years or so. Although I find the subject of the study interesting and highly relevant, I don't find its presentation in the manuscript meets the quality standard, making it suitable for publication in its current form. Therefore, in my opinion, the manuscript should undergo a major revision before being published in Biogeoscienes. I will further explain my reservations in the following:

*Response: We are pleased that the reviewer is considering our manuscript highly relevant and are confident that we are able to present the manuscript in a form meeting the reviewers expectations as well as the quality standard of the journal. Please find the responses to the individual comments below.*

General comments:
1. In the study, forests are combined over the whole globe, providing estimates of the impacts of droughts and/or heatwaves on GPP at a global scale. I wonder, whether it would add value to the study, if also different categories of trees or different climate ranges (which typically also have a dominating type of trees) were distinguished. Different types of trees in a different background climate might be affected by the extreme events in different ways.

*Response: We would like to thank the reviewer for this comment. Differente climate ranges are distinguished in this study so far by using growing season temperature and growing season surface moisture as drivers of the statistical models explaining the impacts of the extreme events. Figure 3 shows the impact of the extreme events using temperature and surface moisture during the event. To further distinguish these impacts in different climate ranges, we added a similar second plot showing the extreme events in climate space opened up by growing season temperature and growing season surface moisture (now: Fig. 3b). We added a paragraph on different climate ranges to the result section (p.10, 176-184).*
*Regarding the second aspect, distinguishing more different categories of trees, we would like to note here, that although forests are combined over the whole globe, the ecosystem type forests provides an astonishing homogeneous response pattern globally buffering negative impacts of extreme events to a certain degree globally. We further differentiated forests into their different land cover classes (such as evergreen needleleaf forest, mixed forest, ...) in Figure 5b. However, splitting different tree categories up to a species level is not possible so far by the means of globally available remote sensing products as used in this study.*

2. I find that the presentation of the results (Section 3) only makes up a relatively small part of the paper, certainly as compared to the introduction and the section on the methodology. I think this section needs to be extended to have a more balanced paper.

*Response: We extended the presentation of the results in a revised version of the paper. Specifically we added a paragraph on different climate ranges (p.10, 176-184), different land cover types (p.11, 209-217) and extended the section on spatial vs. temporal effects (p.11, .220-225).*

3. I find that the conclusions (Section 5) of the paper a bit weak. I think they could be extended in several ways, e.g. what the findings of the study mean for the terrestrial carbon budget and carbon dioxide concentrations under climate change.

*Response: We added a paragraph on our findings with respect to the terrestrial carbon cycle (p.17, 310-315).*

4. I am a bit confused that some of the dots in Fig. 1 seem to be assigned to different types of ecosystems. Unless this is related to the way of presentation, it needs to be explained that grid points can comprehend different types of ecosystems and that in the analysis all (my assumption) types of ecosystems are included rather than the dominating type. I also wonder, whether, if in fact different types are considered, there should be a lower limit on the extent/fraction of the area covered by each type in a grid point.

*Response: We would like to thank the reviewer for pointing us to this possible source of misunderstanding. We extended the explanation of Fig. 1 (now: Fig. 2, p.6 and p.7) as it encompassed all grind cells affected by the extreme event. In many cases one extreme event affects adjacent grid cells which may be dominated by a different ecosystem type. However, each grid cell (resolution 1/12 degree) has still one dominating ecosystem type.*

5. I miss information on the types of ecosystem that are considered in the study in various places. Actually, it seems the only place, where this information can be obtained, is in Fig 4b. The information could easily be provided in a table in Section 2, where the ecosystems could also be grouped in the three main categories: forest, agriculture and others.

*Response: We provided the requested table in section 2 (methods) (p.3).*

6. I miss a discussion of the limitations and potential biases of the data used in the study. This is only done for the FLUXNET data in the discussion (Section 4).

*Response: We thank the reviewer for this important note. Indeed, we specifically discussed the limitations and potential biases of FLUXCOM-RS data as we consider this for the findings of our study to be particularly important. We added a section to the Discussions on the limitations of temperature and radiation (from ERA5) as well as surface moisture (from GLEAM) (p.16, 275-288).*

Specific comments:
Abstract
7. Page 1, lines 10-11: "On the other hand. . . droughts and heatwaves." – That would actually mean a limitation of the data, which to my understanding hasn't been discussed in the paper.

*Response: We would like to note that the limitations of FLUXCOM-RS are discussed as already mentioned by the reviewer (6., see above). However, we extended the discussion to explicitly mentioning the lack of sensitivity to droughts and heatwaves (p. 16, l 289-299) and we added a more detailed discussion of the limitations of temperature, radiation and surface moisture data as outlined above (p.16, 275-288).*

Introduction

8. General: I would find a short paragraph on the structure of the paper at the end of the introduction really helpful.

*Response: We added a short paragraph on the structure of the paper at the end of the introduction (p.17, 301-307).*

9. Page 2, line 27: "the crucial role of timing" – I assume this refers to the timing of the extreme events. Please clarify.

*Response: Yes, indeed. We clarified it to be "crucial role of timing of the extreme event"*

10. Page 2 line 31: "the least understood aspect" – I wonder whether there is a review paper on this or another suitable reference to support this statement.

*Response: We apologise for this statement being a bit speculative and changed it into "one important aspect".*

11. Page 2, line 39: "in some meteorological. . . in ecological processes" – I am not sure, what this statement means. Please clarify.

*Response: We clarified it as follows: "One option is to use values over some global thresholds to detect extremes e.g. to detect temperatures above 40 degree Celsius and to investigate the associated anomaly in vegetation productivity.*

12. Page 2, lines 95-96: "extreme relative to their expected value" – I am not sure that I understand this. In any case, considering a global absolute threshold would not make much sense, while it would make sense to use locally varying thresholds based on the same percentile, e.g. the 95th percentile, would.

*Response: We fully agree with the reviewer. We changed it into: "Another option is to define extreme events relative to some locally varying threshold, e.g. defined by the 95th percentile of the distribution of the data. Here, we rely on the latter definition, and refine the definition by taking also a joint multivariate distribution of the data with regionally varying thresholds into account." (p.2, 39-41).*

Method
13. General: I think it would be nice to properly introduce the acronyms of the various datasets.

*Response: We added the acronyms of the data sets (p.3, Data).*

14. Page 3, line 55: "ERA5" – I think it need to be mentioned that in ERA5 vegetation doesn't vary but is prescribed via some climatological value. That has an effect on the turbulent energy fluxes at the land surface and, thus, might also affect the near-surface temperature.

*Response: We would like to thank the reviewer for this important comment and mention it in the extended discussion of the data limitations (p.16, 275-288).*

15. Page 3, line 57: "GLEAM model-data integration framework" – It would be interesting to know how and to which extent these data are constrained by observations.

*Response: This is indeed an important aspect. GLEAM is driven by precipitation and microwave satellite observations to estimate soil moisture. Surface net radiation and near surface air temperature (from ERA5) are used to estimate evaporation. We mention it in the discussion (p.16, 275-288).*

16. Page 3: line 62: "2003-2018 period" – The choice of this particular time period for the study is not motivated at all.

*Response: This choice represents the common time period of all data sets used. It is mainly constrained by GLEAM v3.3.b (starting 2003, ending 2018) and FLUXCOM-RS (starting 2001, ending 2018). We added the following sentence: "The time period is chosen as it represents the common period of all data sets used at the time of the analysis."*

17. Page 3, line 71: "for more details see the B" – It is not clear, what this means and what it refers to. Appendix B, maybe (see also my comment below)?

*Response: We integrated the appendix into the main text of the paper (p.5, Section 2.3 and Figure 1).*

Results
18. Page 6, line 115: "non-forested land-cover types" – This is one of the (many) places, where information on the types of ecosystems is missing. See also my comment above.

*Response: We specified the non-forested land-cover types ("savannas, grasslands, open and closed shrublands, permanent wetlands") as well as the agriculture land cover type ("C3 and C4*

*croplands as well as C3 and C4 fractions croplands / natural vegetation mosaics")* (p.9/10, 160-163).

19. Page 8, lines 136-137: "the most important. . . model" – I find it interesting to note that according to this statistical model soil moisture doesn't seem to play a role. This is, however, in contrast to the results presented in Fig. 4b, where soil moisture receives a rather large weight. I wonder, how these – at first sight – contrasting results can be reconciled.

*Response: We thank the reviewer to point us to this important aspect. We apologise that this aspect can be misunderstood. We do not state nor do we want to state that soil moisture does not play a role. Soil moisture is one important variable in the statistical model, which we definitely should mention. We mention now, that surface moisture ist the fourth most important variable after land cover type, as can be seen from Fig. 5(a).. Furthermore, we will tone don the first sentence of the paragraph to "Figure 4(a) shows that temperature and soil moisture have some effect on the direction of the impact, but does not consider other potentially important variables. Thus, we refine our understanding of the observed patterns using a statistical model." (p.10 190-191)*

20. Page 8, lines 148-149: "but enhanced productivity. . . contrasting anomalies)" – I am not sure what this statement means. Please clarify.

*Response: We reformulated the statement and extended the paragraph: "… (spatial contrasting anomalies). Apart from an extreme event simultaneously affecting adjacent ecosystems with different or even contrasting impacts, it is also possible that one ecosystem shows contrasting impacts over time. During startup of the extreme event enhanced productivity may be observed which can turn into a contrasting reduced productivity at a later stage of the extreme event. This temporal difference in the response with a longer lasting extreme event is considered to be a temporally contrasting anomaly. To explicitly quantify … (p.11, 220-225)*

Discussion
21. General: I think it would be important to also discuss the potential implications of the effects of extremes on net ecosystem productivity (NEP), given the effects on GPP, to the extent possible.

*Response: We added a paragraph on the implications for net ecosystem exchange on p. 16, 267-274.*

Conclusions
22. General: I think the conclusions need to fill more than the one short paragraph (see my comment above). I also wonder, whether it would be helpful with a short summary of the main results of the study.

*Response: We added a paragraph with a short summary of the main results to the conclusions (p.17 301-307) and added a paragraph on our findings with respect to the terrestrial carbon cycle as stated above (p.17, 310-315).*

Appendix
23. General: I find the appendix unnecessary. This is because, in my view, Fig. A1 should be part of the section on the results (it is discussed quite a bit and is needed to give a complete picture) and Fig. B1 doesn't provide much relevant information (and is not really referred to).

*Response: We integrated the appendix in the main text: we included Figure A1 into the result section as requested (now. Fig. 2c) and we included the extension of the methods into the method section of the paper (p.5, Section 2.3 and Figure 1).*

Figures
24. Figure 1: One of the prominent extreme events ("Russia 2010") is not linked to a dot in the figure. Is this a mistake or doesn't exist a particular grid point that can be assigned to this event?

*Response: We apologise that the linking line of Russia 2020 is hidden behind "Siberia 2011" at the very beginning. We ensured the the link is now visible.*

Also, I think this figure should be extended with the panel representing "other ecosystems", now Fig. A1 in the appendix.

*Response: We moved Figure A1 ("other ecosystems") to the results section (now. Fig. 2c).*

Supplementary material
25. General: An introduction into the structure of the figures, i.e. what the different panels show and how they relate to each other. Also, I think it would be helpful to give the "identification" of the extreme period and the type of extreme (drought, heat wave or a compound even) in a headline. I understand the rational for presenting mean values for temperature and soil moisture, but presenting anomalies instead might highlight some of the regional details and would indicate the soil moisture/temperature coupling. Also, an indication of the colours/numbers of the different ecosystem types shown in the figures would be helpful. That could also be part of the introduction to the supplementary material. See also my comment above.

*Response: We will revise the Supplementary material. Specifically, we will add a general introduction for the structure of the figures and we will add the type of the extreme in a headline. However, we would like to present the figures with mean values as they currently are. The rationale behind presenting mean values instead of relative anomalies is to illustrate the range of global temperatures and surface moisture during extreme events (which are already detected by a relative approach).*

**Response on the**

**Interactive comment by Anonymous Referee 2**

Reviewer: The paper investigates the importance of land cover type in controlling the impacts of climate extremes relative to other factors using a global upscaled product of GPP. The results show that heat and drought events seem to reduce GPP in grasslands and agricultural areas and to increase GPP in forests. The work calls for considering different land cover types in the assessments of the impact of climate extremes on ecosystem functioning. Overall, the objectives of the paper are clear. However, some methodology and results still need further improvement, and some Figure needs to do some improved. I would recommend a major revision. Detailed comments are listed below:

*Response: We would like to thank the author for the feedback on our manuscript. We address the comments in the following more specifically.*

Reviewer: 1. Figure 1 is not intuitive enough; it needs some improvement. It should label the specific events name rather than region and year.

*Response: We would like to thank the reviewer for this comment. As the space in the figure itself is limited, we would prefer to add specific names rather to the caption of the figure, than to the figure itself. However, we would like to note that some of the events have a well known name (e.g. Russian Heatwave 2010, Amazon Drought 2010, European Heatwave 2003, ...) but some do not have well known or clearly defined names (e.g. Siberia 2011, Horn of Africa 2009). We added the specific names to Figure 2.*

Reviewer: 2. I suggest Figure 2 need to label the proportion value.

*Response: We fully agree with the reviewer and would like to thank the reviewer for this suggestion. We now provide labels for the proportions in a revised version of the manuscript (now: Fig. 3).*

Reviewer: 3. Figure 3a is too orderless. I suggest it needs not to label the events.

*Response: We removed the event specific labels from the figure as requested (now: Fig 4a).*

*Reviewer: 4. The authors group land cover classes in forest and agricultural ecosystems, what about grasslands? Abstract illustrates GPP in grasslands is generally reduced during heat and drought events. And which year the land cover data is?*

Response: We will add more details about grasslands to the result section. Figure 5(b) shows that they have a general negative response coefficient in the impact model. We added a paragraph on different land cover types including grasslands (p.11, 208-217)

*Reviewer: 5. I am so fusing about the methodology; I suggest to introduce more detailed of the method about preprocessing and anomaly detection.*

*Response: We added more details about preprocessing and anomaly detection in the method section as requested (p.5/6, 88-108, and section 2.3).*

*Reviewer: 6. The results section needs further analysis, especially need quantitative analysis.*

*Response: We would like to note that the result section provides quantitative statistics on which our findings are based. For instance, we provide fractions of the events with reduced / enhanced productivity including estimates of uncertainty, or we identify the main drivers of the ecosystems response based on gradient boosting machines. We would be very pleased if the reviewer could provide more details on which quantitative analysis he is specifically aiming for. Please note, that we already extended the presentation of the results in the revised version of the paper. Specifically we added a paragraph on different climate ranges (p.10, 176-184), different land cover types (p.11, 209-217) and extended the section on spatial vs. temporal effects (p.11, .220-225).*

[revised manuscript text omitted]

---

## Referee Report (RR1)

*Vegetation modulates the impact of climate extremes on gross primary production* by M. Flach et al.

The study deals with the role of vegetation for the effects of climate extremes on gross primary production (GPP). This is by analysing a selection of different observational data sets for the last 15 years or so. I find the subject of the study interesting and highly relevant. The authors have responded the majority of my comments and suggestions, and I find the manuscript very much improved. Therefore, I now want to recommend the manuscript for publication in Biogeoscienes after minor revision. I will further explain my reservations in the following:

**General comments:**

1. I still find that the presentation of the results (Section 3) makes up a relatively small part of the paper as compared to the introduction and the section on the methodology. The section has been extended somewhat, but maybe there is still some potential for discussing some of the figures in further detail. Also, it might be interesting to relate more to the cases described in the Supplementary Material, where relevant.
2. I am aware that the second reviewer asked for more details on the methodology (Section 2). I am, however, not sure that these additional details are necessary, given that they can be found in the publications that the authors refer to. This means that this section now fills quite a bit.

**Specific comments:**

**Abstract**

3. Page 1, lines 9-10: "On the one hand… has occurred." – I don't really understand the point here. Please clarify.

**Introduction**

4. Page 2 line 36: "other factors" – I think it would be interesting to have some examples of such factors here.
5. Page 2, line 42: I think it should be "can also be considered" here.

**Method**

6. Page 5, lines 102-103: "5% is… (McPhillips et al., 2018)" – This seems to be redundant (see page 4, lines 86-88). Can one of the statements be omitted/shortened?
7. Page 5, caption of Fig. 1: I wonder, whether one could say a bit more about the colour scale, given that readers may not be familiar with it.
8. Page 8, line 133: I think it would be relevant to know, who the regional "growing seasons" have been defined.
9. Page 9: line 150: I wonder, whether the reader needs to know what "bag fraction" means. Is such specific information needed?

**Results**

10. Page 9, caption of Fig. 4: As for part c), it might be helpful to use something like "25% (stippled lines) and 50% (solid lines)".
11. Page 9, lines 158-159: "Our analysis… 2003 and 2018." – This statement was already mode before. Is it necessary to have it two times? If not, it should be removed, where least relevant.
12. Page 10, line 165: "95% confidence interval" – I think it would be better to have that the first time, when the interval is specified, i.e. (56-75%, 95% confidence interval)".
13. Page 10, lines 170-172: "Note that… in this paragraph." – I think a further discussion of Fig. 3 would be very helpful here.
14. Page 10, line 188: "(Figure 4(b))" – I am puzzled by the fact that in the preceding statement different ecosystem types are mentioned, but they cannot be inferred from Fig. 4(b). Is there a mistake?
15. Page 11, line 195: There are two "are", one needs to be omitted.
16. Page 12, line 230: It should be "both in space and time" here.

Discussion

17. Page 15, lines 239-240: It should be "growing seasons are firstly" here.

Conclusions

18. Page 17, line 318: "would expect it to be" – I find this statement very speculative. What is the basis of such an expectation?

References

19. Page 23, line 516: It should be "spatiotemporal" here. I discovered this typo by chance, maybe there are more in the list of references. Please check.

Figures

20. Figure 5: It might be helpful to graphically separate the upper part of b), i.e. the land cover types, from the lower part, representing various other impacts on GPP. Maybe a line, separating the two parts.

Supplementary material

21. Page 3, Siberian heatwave 2011: "in forest (other) ecosystems (43 TgC)… other ecosystems" – I don't understand this statement. Please clarify.

---

## Author Response (AR2)

*Dear Editor,*

*Please find attached the latest version of our manuscript*
*As suggested, we*

- *moved the list of methodological steps in the appendix*
- *improved the contrasts of Figure 3*
- *start the results with a short summary to be readable as a standalone section*
- *improved additional details as requested by the reviewer*

*Unfortunately we currently do not have a native speaker at hand. However, we carefully checked the manuscript again and we will draw on the English language copy editing which is offered by biogeosciences for final revised papers.*

*Please find below a point to point response to the reviewer.*

*Yours sincerely,*
*Milan Flach (on behalf of all co-authors)*
* * *
*Response to the review:*

Vegetation modulates the impact of climate extremes on gross primary production by M. Flach et al.

The study deals with the role of vegetation for the effects of climate extremes on gross primary production (GPP). This is by analysing a selection of different observational data sets for the last 15 years or so. I find the subject of the study interesting and highly relevant. The authors have responded the majority of my comments and suggestions, and I find the manuscript very much improved. Therefore, I now want to recommend the manuscript for publication in Biogeoscienes after minor revision. I will further explain my reservations in the following:

*Response: We would like to thank the reviewer for the positive evaluation and for helping to improve our manuscript.*

General comments:

1. I still find that the presentation of the results (Section 3) makes up a relatively small part of the paper as compared to the introduction and the section on the methodology. The section has been extended somewhat, but maybe there is still some potential for discussing some of the figures in further detail. Also, it might be interesting to relate more to the cases described in the Supplementary Material, where relevant.

*Response: As suggested by the editor, we extend the results by adding a summarising introductory section to improve the readability of the paper as well as we shorten the methods by moving the list of methodological steps into the appendix.*

2. I am aware that the second reviewer asked for more details on the methodology (Section 2). I am, however, not sure that these additional details are necessary, given that they can be found in the publications that the authors refer to. This means that this section now fills quite a bit.

*Response: We agree with the reviewer, that the level of details in the methods is very high now. Therefore, we move the list of methodological steps in the appendix as requested also by the editor.*

Specific comments:

Abstract

3. Page 1, lines 9-10: "On the one hand... has occurred." – I don't really understand the point here. Please clarify.

*Response: We would like to thank the reviewer for this point. We reformulate it to be: "On the one hand, normal to increased GPP values are in many cases plausible, e.g. when conditions prior to the event have been particularly positive. On the other hand, however, normal to increased GPP values in forests may also reflect a lack of sensitivity in current remote sensing derived GPP products to the effects of droughts and heatwaves."*

Introduction

4. Page 2 line 36: "other factors" – I think it would be interesting to have some examples of such factors here.

*Response: We added examples of factors here: "…such as duration and magnitude of the extreme event."*

5. Page 2, line 42: I think it should be "can also be considered" here

*Response: We changed it as requested.*

Method

6. Page 5, lines 102-103: "5% is... (McPhillips et al., 2018)" – This seems to be redundant (see page 4, lines 86-88). Can one of the statements be omitted/shortened?

*Response: We removed the redundant statement.*

7.  Page 5, caption of Fig. 1: I wonder, whether one could say a bit more about the colour scale, given that readers may not be familiar with it.

*Response: We added a sentence to explain the colour scale.*

8.  Page 8, line 133: I think it would be relevant to know, how the regional "growing seasons" have been defined.

*Response: We clarified the definition of the growing season as follows: " Growing season is defined here to be an extreme event taking place in the half year of the GPP maximum (plus/ minus 3 month) "*

9.  Page 9: line 150: I wonder, whether the reader needs to know what "bag fraction" means. Is such specific information needed?

*Response: This specific information is not necessarily needed. We removed it.*

Results

10. Page 9, caption of Fig. 4: As for part c), it might be helpful to use something like "25% (stippled lines) and 50% (solid lines)".

*Response: We added the specification of the lines to the caption of the figure as requested.*

11. Page 9, lines 158-159: "Our analysis... 2003 and 2018." – This statement was already mode before. Is it necessary to have it two times? If not, it should be removed, where least relevant.

*Response: We would like to thank the reviewer for this comment. However, as requested by the editor, we added an introductory paragraph to the results, so that they can be read as standalone section. This means, that some central information which can also be found in the methods is repeated at this place.*

12. Page 10, line 165: "95% confidence interval" – I think it would be better to have that the first time, when the interval is specified, i.e. (56-75%, 95% confidence interval)".

*Response: We changed it as requested.*

13. Page 10, lines 170-172: "Note that... in this paragraph." – I think a further discussion of Fig. 3 would be very helpful here.

*Response: We added a short discussion of Figure 3 as requested: " Thus, Figure 3 also indicates that it does not matter whether we obtain the statistics on an event basis or on the basis of a space-time volume. For both perspectives the main message is the same: agricultural and other ecosystems are most strongly affected by droughts and heatwaves,*

*whereas forests show neutral to enhanced productivity in the majority of the cases.*"

14. Page 10, line 188: "(Figure 4(b))" – I am puzzled by the fact that in the preceding statement different ecosystem types are mentioned, but they cannot be inferred from Fig. 4(b). Is there a mistake?

*Response: We thank the reviewer for this comment. It was indeed a mistake and now refers correctly to Figure 4c.*

15. Page 11, line 195: There are two "are", one needs to be omitted.

*Response: We omitted the second "are".*

16. Page 12, l230: It should be "both in space and time" here.

*Response: We changed it accordingly.*

Discussion

17. Page 15, lines 239-240: It should be "growing seasons are firstly" here.

*Response: We changed it accordingly.*

Conclusions

18. Page 17, line 318: "would expect it to be" – I find this statement very speculative. What is the basis of such an expectation?

*Response: The basis of such statement is, that generally forest are expected to be vulnerable to drought and heat related mortality risks (see e.g. Allen et al., Forest Ecology and Management, 2010). Thus, we would have expected also a more direct negative response of forests to droughts and heatwaves. We modified the statement to clarify: "However, the lack of sensitivity of forest ecosystems to droughts and heatwaves is stronger than we would expect it to be, as forests are generally considered to be vulnerable to drought and heat related mortality risks."*

References:

19 Page 23, line 516: It should be "spatiotemporal" here. I discovered this typo by chance. maybe there are more in the list of references. Please check.

*Response: We corrected the typo. We carefully checked the list of literature and corrected one other mistake in the list of literature.*

Figures

20. Figure 5: It might be helpful to graphically separate the upper part of b), i.e. the land cover types, from the lower part, representing various other impacts on GPP. Maybe a line, separating the two parts.

*Response: We would like to thank the reviewer for this comment. However, we think that separating a figure into (a) and (b) is enough for separating two figures and that no additional line is necessary.*

Supplementary material

21. Page 3, Siberian heatwave 2011: "in forest (other) ecosystems (43 TgC)... other ecosystems" – I don't understand this statement. Please clarify.

*Response: We removed the second "other ecosystems" to clarify.*

[revised manuscript text omitted]